# The Effect of Unhealthy Food Packaging Information Boundaries on Consumer Purchasing Intentions

**DOI:** 10.3390/foods13152320

**Published:** 2024-07-23

**Authors:** Shichang Liang, Junyan He, Wanshan Deng, Ping Cao, Lili Teng, Yu Tang, Xiaojie Lu, Feng Hu, Tingting Zhang, Jingyi Li

**Affiliations:** 1School of Business, Guangxi University, Nanning 530004, China; liangshch@gxu.edu.cn (S.L.); 2329391014@st.gxu.edu.cn (J.H.); caoping@gxu.edu.cn (P.C.); tengll@gxu.edu.cn (L.T.); 20160222@gxu.edu.cn (Y.T.); grzhf70@gxu.edu.cn (F.H.); 18377060952@163.com (T.Z.); 222930101014@st.gxu.edu.cn (J.L.); 2Talents Service Center of Guangxi Zhuang Autonomous Region, Nanning 530004, China

**Keywords:** unhealthy foods, packaging information, boundaries, feeling of constraint, self-construal

## Abstract

Existing studies have examined unhealthy food packaging information, mainly focusing on aspects such as the content, color, and text, whilst paying less attention to the boundaries of information. This paper investigates unhealthy foods through three experiments, revealing that the presence (vs. absence) of packaging information boundaries on unhealthy foods has a negative impact on consumers’ purchasing intentions (*p* = 0.040) (Experiment 1). The feeling of constraint mediates this effect (β = −0.078, CI: [−0.1911, −0.0111]) (Experiment 2). Additionally, consumers with an independent self-construal exhibit reduced purchasing intentions when unhealthy food packaging information boundaries are present (vs. absent) (*p* < 0.001), whereas those with an interdependent self-construal show increased purchasing intentions under the same conditions (*p* = 0.024) (Experiment 3). This paper reveals the psychological mechanism and boundary conditions of unhealthy food packaging information boundaries affecting consumers’ purchasing intention and provides practical inspiration for government policy-making related to unhealthy food packaging.

## 1. Introduction

In recent years, global concern regarding unhealthy diets and obesity has significantly increased [1]. A contributing factor to obesity is the frequent consumption of unhealthy foods, which are typically high in sugar, calories, and fat, as individuals seek to satisfy the reward circuits in their brains [2]. It is also a key reason why people choose unhealthy food; it is both rewarding and enjoyable [3]. Individuals who are strongly attracted to unhealthy foods are more prone to making less healthy dietary choices and gaining more weight [4]. Consequently, numerous scholars have investigated the reasons for consuming unhealthy foods. Research has indicated that positive emotions can lead to increased consumption of high-calorie foods [5]. Individuals experiencing sadness are more likely to consume unhealthy foods [6]. Additionally, economic factors also play a role; those with low incomes often deliberately choose unhealthy foods that are high in fat to meet their caloric needs [7]. Individuals residing in disadvantaged neighborhoods face limited access to quality food, greater exposure to unhealthy snacks, and snack advertisements [8].

Several scholars have also explored strategies to inhibit the consumption of unhealthy foods. For instance, prolonged exposure to the odors of unhealthy foods (exceeding two minutes) has been shown to decrease consumers’ willingness to purchase these items [9]. Additionally, exposure to actual foods or food cues before purchase can reduce the consumption and enjoyment of unhealthy foods [10]. Emphasizing self-control, such as resisting tempting foods, diminishes consumers’ desire to eat or their actual intake [10]. Individuals with higher BMI consider that their future lives will lead to a decrease in the consumption of unhealthy foods [11]. Moreover, digital exposure to unhealthy foods can prompt individuals to consider healthy eating, thereby reducing their consumption of unhealthy foods [12], etc. 

Much of the research on unhealthy food packaging has concentrated on the information conveyed by the packaging itself. For example, a product in a glossy package is perceived to be less healthy [13]. Blue or green packaging typically indicates healthy foods, whereas red packaging usually signifies tasty but less healthy alternatives [14]. Foods with shorter brand names are generally perceived as healthier [15]. The main disincentive for unhealthy food packaging is nutritional labeling. The measure has been adopted and put into effect in various countries including China, Greece, the United States, Switzerland, Spain, Iran, and Ecuador [16]. Existing studies have primarily concentrated on the influence of individual consumer factors or the impact of the consumption environment on consumer behavior. In research specifically examining unhealthy food packaging, the emphasis has predominantly been on the content of the information, such as color [14] and text [15], with insufficient attention given to the boundaries of the unhealthy food packaging information. Consequently, this paper aims to address this gap by exploring the impact of the presence (vs. absence) of unhealthy food packaging information boundaries on consumers’ purchasing intentions. 

The term “boundary” refers to the border of an object [17]. Boundaries play a crucial role in identifying objects within a given space and symbolize structure and order [18]. Research in the field of aesthetics suggests that design elements possess both embodied meaning and referential meaning [19]. Embodied meaning pertains to the inherent and essential significance of the elements themselves, such as information boundaries that allow for the separation of information [20,21]; referential meaning pertains to the associations individuals derive from design elements. For instance, information boundaries in a design evoke associations with spatial boundaries, thereby activating an individual’s perception of spatial constraints [17,22]. When space is bounded, individuals often feel restricted, hindered, and inhibited, resulting in a sense of constraint [23]. The feeling of constraint refers to the feeling of restricted freedom, which is generally an unfavorable experience for individuals [24]. Building on this, we posit that the presence (vs. absence) of unhealthy food packaging information boundaries activates a feeling of constraint. This feeling of constraint, along with the associated feelings of limitation and burden, subsequently reduces individuals’ willingness to purchase unhealthy foods. Furthermore, different types of self-construal generate distinct consumer goals and social orientations, thereby influencing their psychology and decision-making processes [25]. Given this, self-construal moderates the effect of unhealthy food packaging information boundaries on consumers’ purchasing intentions. Specifically, consumers with an independent self-construal exhibit reduced purchasing intentions when unhealthy food packaging information boundaries are present (vs. absent), whereas those with an interdependent self-construal show increased purchasing intentions under the same conditions.

There are three main contributions of this paper. First, it expands the research on unhealthy food packaging information. While the existing literature primarily focuses on aspects such as content, color, and wording, it pays less attention to the boundaries of unhealthy food packaging information. Therefore, this study addresses this gap by exploring the impact of the presence (vs. absence) of unhealthy food packaging information boundaries on consumers’ purchasing intentions. Second, it contributes to the literature on the feeling of constraint in unhealthy food packaging information. The findings reveal that the presence (vs. absence) of boundaries in unhealthy food packaging information triggers a feeling of constraint in individuals when considering the purchase of unhealthy foods. This feeling of constraint induces feelings of limitation and burden, subsequently decreasing individuals’ willingness to purchase such foods. These insights further advance research related to the feeling of constraint in the context of unhealthy food consumption. Finally, this study extends the research on self-construal in the context of unhealthy food consumption. The results indicate that consumers with an independent self-construal exhibit reduced purchasing intentions in the presence (vs. absence) of boundaries in unhealthy food packaging information. Conversely, consumers with an interdependent self-construal show increased purchasing intentions under the same conditions. These findings further enrich the literature on self-construal and its impact on unhealthy food consumption behavior.

## 2. Literature Review and Research Hypothesis

### 2.1. Unhealthy Foods and Packages

Unhealthy foods are characterized by a high caloric content, low nutritional value, and high fat levels. These foods typically contain minimal amounts of fruits and vegetables [26,27]. Common examples include fast food, soft drinks, candies, snacks, processed meats, cakes, ice cream, and fried foods [28]. Additionally, scholars have identified key nutrients and developed various labeling schemes, such as the Swedish Keyhole mark, the front-of-pack traffic light labeling in the United Kingdom, and the nutrient profiling model [29]. These schemes more precisely define unhealthy foods based on parameters such as calories, salt, and sugar [29]. In recent years, there has been a marked increase in global attention to unhealthy diets. Numerous scholars have examined unhealthy foods from various perspectives. Firstly, regarding the causes of unhealthy food consumption, factors such as media advertising campaigns [30], working hours [31], and the proliferation of fast-food restaurants [32] increase individuals’ exposure to unhealthy foods, thereby elevating the consumption of unhealthy foods. Additionally, peer influence is a significant contributing factor [33]. Personal factors also play a role in unhealthy food consumption, including experiences of interpersonal rejection [34], individual stress levels [35], and impulsivity [36], all of which can lead to increased intake of unhealthy foods. Accordingly, scholars have explored various inhibition strategies. For instance, policy–behavioral approaches include banning the promotion of unhealthy products in schools [37] and imposing taxes on unhealthy foods [38]. Additionally, strategies such as encouraging consumers to imagine consuming unhealthy foods [39], displaying unhealthy foods in a cluttered manner [40], and engaging consumers in inhibition games related to unhealthy foods [41] have been shown to reduce consumers’ desire to consume and willingness to purchase these items.

In contrast, research concerning unhealthy food packaging has primarily examined how the healthiness of food can be communicated through various packaging displays. Factors such as the smoothness of the packaging [42], the color scheme employed [14], the length of the brand name [15], the material used for packaging [43], the color of the nutritional label [44], and different visual presentations influence consumers’ perception of the healthiness of the food; with regard to the disincentive strategy on the package itself, some governments have adopted an approach of requiring front-of-package labeling to help consumers make informed purchasing and consumption choices. The Chinese government has made nutritional labeling mandatory for prepackaged foods [45]. This measure has also been implemented in countries such as Greece, the United States, Switzerland, Spain, and so on [16]. Overall, the existing literature on unhealthy food packaging information predominantly centers on aspects such as food content [45], color [14], and wording [15], while comparatively less attention has been directed toward the boundaries of food information.

### 2.2. Boundaries 

The concept of “boundary”, first explicitly introduced by Cutright, refers to the demarcation of an object, distinguishing it from its surroundings and signifying its belonging [17]. Boundaries can be categorized into tangible and intangible forms; tangible boundaries include elements such as picture frames and logo frames, while intangible boundaries are represented through orderly product placements, such as neatly arranged items on shelves [17]. Boundaries are crucial for identifying objects within a given space and serve as presentations of structure and order [18]. In the fields of sociology, psychology, and consumer behavior, scholars have found that boundaries can significantly impact individuals’ cognition. For instance, numbers slightly above a categorical boundary (e.g., 1001 compared to 1000) induce arousal and enhance consumer cravings [46]. Setting boundaries between different dishes on a menu can eliminate the psychological contagion effect, thereby mitigating the negative impact of uncomfortable dishes on other items [20]. When consumers experience a lack of control, they tend to prefer goods with clear boundaries, compensating for their need for control through the acquisition of order and structure [17]. Visual boundaries significantly influence consumers’ perceptual diversity under high cognitive load, and the order in which people process information is affected by these boundaries [47]. Individuals segment their behavior according to the lines demarcating different periods on a calendar, leading them to make decisions independently within each interval [21]. Additionally, boundaries can reduce the impact of brand extensions, weaken brand flexibility, and diminish brand breadth [48].

### 2.3. Unhealthy Food Packages and Information Boundaries

Unhealthy food packages typically display a variety of information. Besides the brand and images on the front, a significant portion of the information is located on the sides and back, including details such as shelf life, storage conditions, and ingredient lists [49]. Food packaging information serves as the sole source of data for consumers at the time of purchase [50]. Consumers interpret and respond differently to various types of information (text, numbers, graphs, etc.) [51]. For example, research indicates that consumers pay close attention to direct quality information like shelf life [52]. Consequently, companies often emphasize this information using different colors [53], fonts [54], and other visual techniques. A border is a visual boundary surrounding a focal object [22]. Adding borders around product information to highlight it spatially is a common practice among merchants. However, the effectiveness of using borders in food information presentation remains questionable. Specifically, does the use of borders influence consumers’ attitudes toward unhealthy foods?

Based on the referential meaning of design elements, individuals can form associations, such as informational boundaries in design reminding people of spatial boundaries [17,22]. Physical architecture embodies spatial boundaries that can convey specific semantic associations; for example, small and enclosed churches can evoke feelings of constraint [55]. When the ceiling height is low, individuals feel restricted, hindered, and inhibited [56]. Similarly, narrow aisles create a feeling of constraint by limiting the space available for movement [23]. The establishment of spatial boundaries constrains the scope of people’s activities, imposing a feeling of constraint and limitation. According to associative network theory [57], when the referential meaning of a design element is activated, related associations are triggered through paths in the associative network. Thus, the boundary of product information can evoke associations with spatial boundaries, activating the corresponding constraining associations. Consequently, the food packaging information boundary, as a design element, can induce a feeling of constraint. This feeling of constraint refers to the feeling of limited freedom, which is generally undesirable [24]. Constrained consumers not only experience increased mental burden but also exhibit reduced cognitive bandwidth in tasks involving reading, thinking, and decision-making [58].

Unhealthy foods, also known as indulgent foods [9], which provide consumers with intense pleasure and hedonic stimulation, offering significant emotional benefits [59]. Essentially, when choosing between healthy and unhealthy foods, consumers are balancing their long-term health goals against their short-term hedonic desires [9]. Consumers with strong self-restraint are more likely to adhere to their long-term goals and choose healthy foods, while those with weaker self-restraint tend to favor indulgent options [60]. In this context, indulgence and restraint can be seen as two sides of the same coin [61]. Thus, when consumers purchase unhealthy foods for indulgence, the feeling of constraint imposed by the food packaging information boundary conflicts with their desire for indulgence. This conflict can elicit stronger negative emotions and further decrease their purchasing intention.

**H1.** 
*The presence (vs. absence) of boundaries in unhealthy food packaging information reduces consumers’ purchasing intentions.*


**H2.** 
*The feeling of constraint mediates the impact of the presence (vs. absence) of boundaries in unhealthy food packaging information on consumers’ purchasing intentions.*


### 2.4. The Moderating Role of Self-Construal

Self-construal refers to the extent to which an individual perceives the self as connected to or separate from others. It defines how an individual views the self about others and is typically categorized into independent and interdependent self-construal [62]. Independent self-construal emphasizes individuality and personal achievement [63] and is closely associated with Western culture [60]. In contrast, interdependent self-construal focuses on relationships and commonalities with others [64] and is more closely linked to Eastern culture [60]. In addition, gender influences the type of individual self-construal, with males typically exhibiting more independent self-construal than females [65]. Some researchers argue that individuals can have both an independent and an interdependent self-construal, and one type may dominate in a given situation; different manipulation methods can be used to initiate different types of self-construal in individuals in the short term [66,67]. Previous studies have demonstrated that self-construal can influence consumers’ attitudes towards products [68] and affect the relative attention consumers pay to themselves and others when making decisions [67]. Subsequent research has expanded on these findings, exploring areas such as brand strategy, advertising appeals [69], and word-of-mouth communication [70], etc. Additionally, different types of self-construal induce distinct consumer goals and social orientations, impacting their psychology and decision-making processes [25].

Past research has shown that individuals with an independent self-construal tend to emphasize their motivations, goals, attitudes, internal feelings, and behaviors [71]. This tendency to focus on their own internal attributes or characteristics promotes the pursuit of personal interests [72]. They also rely more on their internal feelings to shape their consumption decisions [73]. Consequently, when individuals with an independent self-construal seek to purchase unhealthy foods, the presence of boundaries in unhealthy food packaging information can evoke feelings of constraint and limitation. This is incongruent with their psychological motivation for indulgence, thereby reducing their willingness to purchase.

In contrast, an interdependent self-construal places greater emphasis on the relationship between the self and others [74] and on social connectedness [62]. Individuals with an interdependent self-construal tend to think and interact within groups or strengthen existing relationships [75], emphasizing harmonious social values [76]. Their focus is typically on enhancing the collective welfare of society [62], fostering social cohesion, and adhering to social norms [77]. Overall, interdependent individuals not only enhance their perception of kinship with social goals but also engage in more holistic thinking [78]. Therefore, we hypothesize that interdependent individuals will increase their willingness to purchase unhealthy foods because the feeling of constraint brought about by boundaries aligns with their need for group conformity and social norms.

**H3.** 
*Self-construal moderates the effect of boundaries in unhealthy food packaging information on purchasing intentions.*


**H3a.** 
*Consumers with an independent self-construal exhibit reduced purchasing intentions when boundaries in unhealthy food packaging information are present (vs. absent).*


**H3b.** 
*Consumers with an interdependent self-construal show increased purchasing intentions when boundaries in unhealthy food packaging information are present (vs. absent).*


The overall conceptual model is depicted in Figure 1.

## 3. Experiment 1: Validating the Effect of Unhealthy Food Packaging Information Boundaries on Purchasing Intentions

### 3.1. Experimental Purpose

Experiment 1 utilized a between-subjects design with a single factor (boundary of unhealthy food packaging information: presence vs. absence). The aim was to examine the effect of the boundary of unhealthy food packaging information on purchasing intention. We hypothesized that consumers’ willingness to purchase unhealthy foods would decrease when the information on the packaging is bounded.

### 3.2. Experimental Sample and Design

We recruited participants for this experiment through the Credamo platform (a well-known intelligent professional research platform in China), resulting in a final sample of 161 participants (M_age_ = 30.44 years, SD = 9.60, 61.6% female). The entire experimental process was conducted online without any researcher assistance. After the experiment, each participant was randomly paid 5–10 yuan.

To ensure ecological validity, a real-world brand was utilized as the experimental stimulus. Additionally, the unhealthy foods used had been identified by past research, and the product information displayed was true information about the product, which included the product name, net weight, ingredients, shelf life, etc. 

### 3.3. Experimental Procedure

The experiment included two parts: viewing stimuli and answering questions.

Participants were randomly assigned to one of two groups, the boundary-presence group, or the boundary-absence group. Participants in the boundary-presence group saw the unhealthy food packaging information enclosed by a frame, while participants in the boundary-absence group saw the unhealthy food packaging information without a frame. The content of the food information was identical for both groups The boundary in this experiment was implemented using a rectangular frame.

Then, participants were asked to imagine that they were on a fat-reducing diet, but after a busy workday, planned to reward themselves by buying a hamburger after work. At this point, participants viewed an image of the product information for Sam’s hamburger [28] and were informed that this was a screenshot of the product information on the hamburger’s packaging (see Figure 2). Participants were required to remain in this stage for 1 min to carefully observe the stimulus.

Next, to test the success of the boundary manipulation, participants were asked to respond to the item “This product information has a frame”, using a 7-point Likert scale (1 = strongly disagree, 7 = strongly agree). 

Finally, the measurement of purchasing intention was conducted using a 7-point Likert scale comprising two items, “I would consider purchasing this hamburger”, and “I might purchase this hamburger” [79]. Also, measures for perceived product quality [80] and perceived innovativeness [81] were administered, along with demographic information and an attention check question. (See Appendix A for scales and Appendix B for an overview of experiments).

**Figure 2 foods-13-02320-f002:**
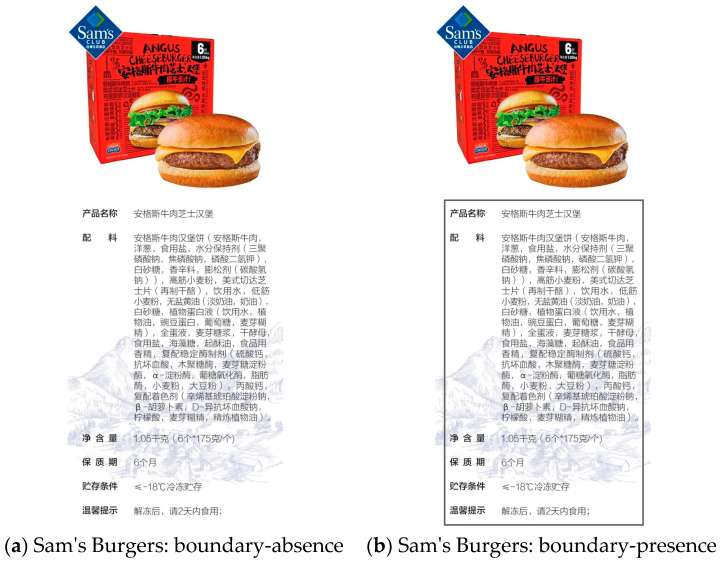
Stimuli materials of Experiment 1. (**a**) Sam’s Burgers: boundary-absence. (**b**) Sam’s Burgers: boundary-presence. The content of the food information was identical for both figures. The Chinese meaning of the above diagram is as follows.


Product name: Angus Beef Cheese BurgerIngredients List: Angus Beef Patty (Angus beef, onion, salt, moisture-retaining agents [sodium tripolyphosphate, sodium pyrophosphate, monopotassium phosphate], white sugar, spices, leavening agent [sodium bicarbonate]), High-Gluten Wheat Flour, American Cheddar Cheese Slices (processed cheese), Drinking Water, Low-Gluten Wheat Flour, Unsalted Butter (light cream, cream), White Sugar, Vegetable Protein Liquid (drinking water, vegetable oil, pea protein, glucose, maltodextrin), Whole Egg Liquid, Malt Syrup, Dry Yeast, Salt, Trehalose, Shortening, Food Flavoring, Compound Stabilizing Enzyme Preparation (calcium sulfate, ascorbic acid, xylanase, maltogenic amylase, α-amylase, glucose oxidase, lipase, wheat flour, soybean flour), Calcium Propionate, Compound Coloring Agent (sodium starch octenyl succinate, β-carotene, sodium erythorbate, citric acid, maltodextrin, refined vegetable oil)Net content: 1.05 kg (6 pieces * 175 g/piece)Shelf life: 6 monthsStorage condition: <−18 °C frozen storageReminder: After thawing, please consume within 2 days


### 3.4. Results

Manipulation test: An analysis of variance (ANOVA) was conducted with the presence of a boundary around unhealthy food packaging information as the dependent variable. The results showed that the participants in the boundary-presence group perceived a significantly higher extent of boundary presence (M _presence_ = 6.45, SD _presence_ = 0.80) compared to those in the boundary-absence group (M _absence_ = 3.11, SD _absence_ = 1.88; F (1, 159) = 208.59, *p* < 0.001, η^2^ = 0.57). 

Main effects: An analysis of variance (ANOVA) was conducted with the purchasing intention as the dependent variable. The results showed that the participants in the boundary-presence group (M _presence_ = 4.90, SD _presence_ = 1.34) had a significantly lower purchasing intention compared to those in the boundary-absence group (M _absence_ = 5.30, SD _absence_ = 1.12; F (1, 159) = 4.28, *p* = 0.040, η^2^ = 0.26). This indicates that the boundary around the unhealthy food packaging information reduced the purchasing intention, supporting H1 (see Figure 3).

Alternative explanations: We conducted an analysis of variance (ANOVA) with perceived quality and perceived innovativeness as the dependent variables and the presence or absence of boundaries in unhealthy food packaging information as the independent variable. The results indicated that the participants’ perceived quality (M _absence_ = 5.44, SD _absence_ = 1.17 vs. M _presence_ = 5.27, SD _presence_ = 1.01; F (1, 159) = 0.95, *p* = 0.332, η^2^ = 0.01) and perceived innovativeness (M _absence_ = 5.15, SD _absence_ = 1.24 vs. M _presence_ = 5.40, SD _presence_ = 1.07; F (1, 159) = 1.84, *p* = 0.177, η^2^ = 0.01) were not affected by the boundary of the unhealthy food packaging information. In addition, we employed 5000 bootstrap samples to perform a mediation analysis on perceived quality and perceived innovativeness [82]. The indirect effect interval of perceived quality included 0 (β = −0.1264, SE = 0.1316, 95% CI = [−0.3936, 0.1230]), and for perceived innovativeness, the indirect effect interval also included 0 (β = 0.1095, SE = 0.0847, 95% CI = [−0.0472, 0.2847]). This observation suggests that perceived quality and perceived innovativeness are not potential explanatory factors.

### 3.5. Discussion

Experiment 1 examined the relationship between the boundaries of unhealthy food packaging information and consumers’ purchasing intentions, providing empirical support for Hypothesis 1. As expected, compared to the absence of a boundary on unhealthy food packaging information, the participants perceived that information with the presence of a boundary was less likely to incite purchasing intentions. At the same time, Experiment 1 also excluded the effects of perceived quality and perceived innovativeness on the findings of the study. However, this initial exploration has some shortcomings. Firstly, the experiment selected hamburgers as the stimulus, which might have influenced purchasing intentions due to participants’ potential satiety needs. Future studies should use more neutral food items to verify the findings. Additionally, whether the shape of the boundary affects the main effect remains to be explored. Finally, the mechanism through which food information boundaries influence purchasing intentions also requires further investigation.

## 4. Experiment 2: The Mediating Effect of Feeling of Constraint on the Relationship between Unhealthy Food Packaging Information Boundaries and Purchasing Intentions

### 4.1. Experimental Purpose

Experiment 2 aimed to provide an underlying explanatory mechanism for the effect of unhealthy food packaging information boundaries on purchasing intention. The experiment employed a between-subjects design with a single factor (boundary of unhealthy food packaging information: presence vs. absence). We hypothesized that the presence of a boundary around unhealthy food information would induce a feeling of constraint, thereby reducing the intention to purchase unhealthy foods.

### 4.2. Experimental Sample and Design

We recruited participants for this experiment through the Credamo platform (a well-known intelligent professional research platform in China), resulting in a final sample of 217 participants (M_age_ = 31.42 years, SD = 9.08, 70.04% female). The entire experimental process was conducted online. The entire experimental process was conducted online without any researcher assistance. After the experiment, each participant was randomly paid 5–10 yuan.

To ensure the generalizability of findings, Experiment 2 used a circular frame. In addition, we used a more neutral food, i.e., chips [83], to verify the hypothesis. The product information included the product name, net weight, ingredients, production date, and other details. 

### 4.3. Experimental Procedure

Consistent with the previous experiments, participants were randomly assigned to one of two groups, the boundary-presence group or the boundary-absence group.

Then, participants were also asked to imagine a consumption scenario for healthy foods. At this point, participants viewed an image of the product information for chips (see Figure 4) and were informed that this was a screenshot of the product information on the chips’ packaging. Participants were required to remain on this screen for one minute, during which they were to carefully observe the stimulus. After viewing the picture stimulus, participants proceeded to the next screen to answer the questions, where the stimulus image was no longer visible. The manipulation check for the boundary was conducted in the same manner as in Experiment 1. 

Next, participants rated the product’s attractiveness using the items “This is an attractive bag of chips” and “This is a well-liked bag of chips” [84]. In order to test the proposed explanatory mechanism, we conducted assessments of the feeling of constraint; “While reading the product information on the chip packaging, I felt inhibited” and “While reading the product information on the chip packaging, I felt constrained” [58]. All scales were evaluated using a 7-point Likert scale. 

Finally, to exclude potential alternative mechanisms, we utilized a 7-point Likert scale to measure perceptions of warmth, i.e., “I think this is a warm frame” [85] and “I think this frame is friendly” [86], as well as a familiarity scale regarding the product [87]. In addition, participants completed demographic information and an attention check question. (See Appendix A for scales and Appendix B for an overview of experiments).

**Figure 4 foods-13-02320-f004:**
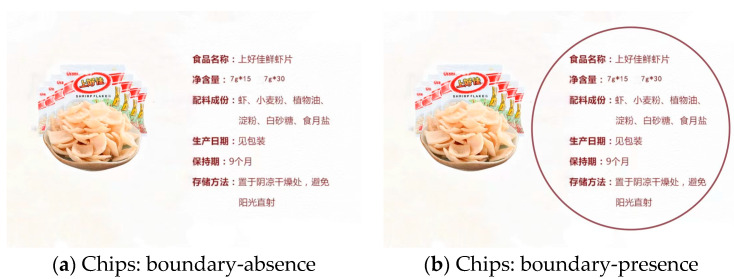
Stimuli materials of Experiment 2. (**a**) Chips: boundary-absence (**b**) Chips: boundary-presence. The content of the food information was identical for both figures. The Chinese meaning of the above diagram is as follows.


Product name: Oishi shrimp flakesNet content: 7 g*15 7 g*30Ingredients List: Shrimp, wheat flour, vegetable oil, starch, white sugar, edible saltDate of manufacture: See packagingShelf life: 9 monthsStorage method: Store in a cool, dry place, away from direct sunlight


### 4.4. Results

Manipulation test: An analysis of variance (ANOVA) was conducted with the presence of a boundary around the unhealthy food packaging information as the dependent variable. The results indicated that compared to the boundary-absence group (M _absence_ = 2.92, SD _absence_ = 2.04), the participants in the boundary-presence group perceived the unhealthy food information as significantly more bounded (M _presence_ = 5.44, SD _presence_ = 1.69; F (1, 215) = 98.22, *p* < 0.001, η^2^ = 0.31). 

Main effects: An analysis of variance (ANOVA) was conducted with the product’s attractiveness as the dependent variable. The results indicated that the participants in the boundary-presence condition found the chips less attractive (M _presence_ = 5.38, SD _presence_ = 1.22) compared to those in the boundary-absence condition (M _absence_ = 5.68, SD _absence_ = 0.79; F (1, 215) = 4.78, *p* = 0.030, η^2^ = 0.02), suggesting that the presence of a boundary in unhealthy food packaging information reduces its attractiveness. These findings provide additional support for Hypothesis 1.

Mediated analysis: An analysis of variance (ANOVA) was conducted with the boundary of the unhealthy food packaging information as the independent variable and perceived constraint as the dependent variable. The results indicated that the participants in the boundary-presence condition (M _presence_ = 3.42, SD _presence_ = 1.64) perceived a higher level of constraint compared to those in the boundary-absence condition (M _absence_ = 2.84, SD _absence_ = 1.54); F (1, 215) = 7.17, *p* = 0.008, η^2^ = 0.03). A mediation analysis was then conducted using the boundary of the unhealthy food packaging information as the independent variable, feeling of constraint as the mediator, and attractiveness as the dependent variable, incorporating the mediation model (Model 4, bootstrapping 5000 times) [82]. The results showed that the mediating effect of feeling of constraint on the relationship between the boundary of the unhealthy food packaging information and product attractiveness was significant (β = −0.078, SE = 0.047, 95% CI: [−0.1911, −0.0111]) (see Figure 5).

Alternative explanations: We conducted an analysis with warmth and familiarity as the dependent variables and the boundary of the unhealthy food packaging information as the independent variable. The results indicated that the participants’ warmth (M _absence_ = 5.57, SD _absence_ = 0. 82 vs. M _presence_ = 5.39, SD _presence_ = 1.08; F (1, 215) = 1.823, *p* = 0.178, η^2^ = 0.01) and familiarity (M _absence_ = 5.93, SD _absence_ = 0.94 vs. M _presence_ = 5.74, SD _presence_ = 1.30; F (1, 215) = 1.73, *p* = 0.190, η^2^ = 0.01) were not significantly affected by the boundary of the unhealthy food packaging information. Additionally, we conducted mediation analyses with 5000 bootstrap samples for warmth and familiarity [82]. The indirect effect interval of warmth included 0 (β = −0.1376, SE = 0.1015, 95% CI = [−0.3251, 0.0656]), and the indirect effect interval of familiarity included 0 (β = −0.0878, SE = 0.0675, 95% CI = [−0.2275, 0.0367]). This observation suggests that warmth and familiarity are not potential explanatory factors.

### 4.5. Discussion

Experiment 2 once again validated the main effect while confirming the establishment of the constraint mechanism, namely, that feeling of constraint is the underlying reason for the reduction in purchasing intention caused by the boundary of unhealthy food packaging information (H2). Additionally, this experiment excluded the potential influences of food familiarity and warmth. Moreover, there were the following limitations: Firstly, the stimuli used in Experiments 1–2 were real brands on the market, and the consumers may have been influenced by familiarity and brand preferences. Secondly, the boundary (frame) can be perceived as protective, and a sense of control may be a potential mediating variable. Finally, it is essential to examine the effects of prices and liking.

## 5. Experiment 3: The Moderating Effect of Self-Construal on the Relationship between Unhealthy Food Packaging Information Boundaries and Purchasing Intentions

### 5.1. Experimental Purpose

Experiment 3 was designed to test Hypothesis 3, which concerns the boundary role of self-construal, and to replicate the fundamental and mediating effects discussed in this paper within the context of virtual branding. The experiment utilized a 2 (boundary of unhealthy food information: presence vs. absence) × 2 (self-construal: interdependent vs. independent) between-subjects design. We hypothesized that consumers with an independent self-construal would exhibit reduced purchasing intentions when boundaries are present (vs. absent) (H3a). Conversely, we hypothesized that consumers with an interdependent self-construal would show increased purchasing intentions when boundaries are present (vs. absent) (H3b).

### 5.2. Experimental Sample and Design

We recruited participants for this experiment through the Credamo platform (a well-known intelligent professional research platform in China), resulting in a final sample of 341 participants (M_age_ = 30.93 years, SD = 7.57, 64.22% female). The entire experimental procedure was conducted online. Consistent with the preceding experiments, the entire procedure was conducted online without researcher intervention. Upon completion, participants were compensated with a randomized payment of 5–10 yuan.

To control for the potential influence of brand familiarity and brand preference, a virtual brand was used in experiment 3. In this experiment, chocolate cookies [88] were used as the stimulus. Participants viewed the food packaging information for the chocolate cookies, which included details such as the product name, ingredients, net weight, shelf life, and nutritional information (see Figure 6).

**Figure 6 foods-13-02320-f006:**
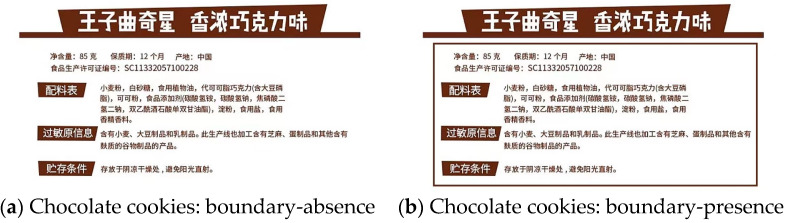
Stimuli materials of Experiment 3. (**a**) Chocolate cookies: boundary-absence. (**b**) Chocolate cookies: boundary-presence. The content of the food information was identical for both figures. The Chinese meaning of the above diagram is as follows.


Net content: 85 gShelf life: 9 monthsOrigin: ChinaFood Production License No. SC11332057100228Ingredients List: Wheat Flour, White Sugar, Edible Vegetable Oil, Cocoa Butter Substitute Chocolate (contains soy lecithin), Cocoa Powder, Food Additives (ammonium bicarbonate, sodium bicarbonate, disodium pyrophosphate, diacetyl tartaric acid esters of mono- and diglycerides), Starch, Edible Salt, Edible Flavorings.Allergen information: Contains wheat, soy products, and dairy products. This production line also processes products containing sesame, egg products, and other grain products containing glutenStorage conditions: Store in a cool and dry place, away from direct sunlight


### 5.3. Experimental Procedure

In this experiment, participants were randomly assigned to one of four groups. 

This study aimed to initiate the subjects’ self-construal using guidance grammar [89]. Participants were given a 3 min time frame to think and write down their answers based on the guiding phrase assigned to them. The independent self-construal group was instructed, “Please think about what is expected of you”. In contrast, the interdependent self-construal group was instructed, “Please think about what is expected of you by your family or friends”.

Then, participants were first asked to imagine that they were on a fat-reducing diet but, after a busy workday, planned to reward themselves by buying a box of chocolate cookies after work. Participants then viewed the product information for the chocolate cookies. Participants were required to remain in this stage for 1 min to carefully observe the stimulus. After viewing the picture stimulus, participants proceeded to the next screen to answer the questions, where the stimulus image was no longer visible.

Next, the procedure started by assessing the effect of the manipulation by asking the participants two questions, i.e., “What I just thought made me think of myself” and “What I just thought made me think of my friends/family” [90], using a 7-point Likert scale. And the manipulation check for the boundary was conducted in the same manner as in Experiment 2.

Then, participants evaluated the product using items such as “I am satisfied with these chocolate cookies” and “I rate these chocolate cookies highly” [91]. And the manipulation check for the boundary was conducted in the same manner as in Experiment 2. 

Finally, using a 7-point Likert scale to rule out potential alternative mechanisms, participants’ sense of control was measured using the item, “While reading the product information, I felt that everything was under my control” [92]. In addition, participants completed measures of perceived price [93] and product liking [94]. Additionally, participants completed demographic information and an attention check question, all on a 7-point Likert scale. (See Appendix A for scales and Appendix B for an overview of experiments).

### 5.4. Results

Manipulation test: An analysis of variance (ANOVA) was conducted with the presence of a boundary around the unhealthy food packaging information as the dependent variable. The results indicated that, compared to the boundary-absence group (M _absence_ = 3.09, SD _absence_ = 1.81), the participants in the boundary-presence group perceived the unhealthy food packaging information as significantly more bounded (M _presence_ = 6.47, SD _presence_ = 0.77; F(1, 336) = 498.42, *p* < 0.001, η^2^ = 0.597). Through analysis of variance (ANOVA), it was found that the participants in the independent self-construal group exhibited a significantly higher tendency towards an independent self-construal (M _independent_ = 4.41, SD _independent_ = 0.87) than those in the interdependent self-construal group (M _interdependent_ = 3.35, SD _interdependent_ = 1.02; F (1, 335) = 107.00, *p* < 0.001, η^2^ = 0.24). Conversely, the participants in the interdependent self-construal group exhibited a significantly higher tendency towards an interdependent self-construal (M _interdependent_ = 4.65, SD _interdependent_ = 1.02) than those in the independent self-construal group (M _independent_ = 3.59, SD _independent_ = 0.87, F (1, 335) = 107.00, *p* < 0.001, η^2^ = 0.24). These results indicate that the manipulation of self-construal was successful.

Main effects: An analysis of variance (ANOVA) was conducted with the product evaluation as the dependent variable. The results indicated that the participants in the boundary-presence condition (M _presence_ = 4.99, SD _presence_ = 1.22) rated the chocolate cookies lower compared to those in the boundary-absence condition (M _absence_ = 5.27, SD _absence_ = 1.14; F (1, 336) = 4.95, *p* = 0.027, η^2^ = 0.015). This finding suggests that the presence of boundaries in the unhealthy food packaging information decreased the product evaluation of the chocolate cookies, providing further support for Hypothesis 1.

Mediated analysis: An analysis of variance was conducted with the boundary of the unhealthy food packaging information as the independent variable and feeling of constraint as the dependent variable. The results indicated that the participants in the boundary-presence condition (M _presence_ = 3.35, SD _presence_ = 1.65) felt a higher feeling of constraint compared to those in the boundary-absence condition (M _absence_ = 2.77, SD _absence_ = 1.52; F (1, 336) = 11.29, *p* < 0.001, η^2^ = 0.032). A mediation analysis was then conducted using the boundary of the unhealthy food packaging information as the independent variable, feeling of constraint as the mediator, and product evaluation as the dependent variable, incorporating the mediation model (Model 4, bootstrapping 5000 times) [82]. The results showed that the mediating effect of feeling of constraint on the relationship between the boundary of the unhealthy food packaging information and product evaluation was significant (β = −0.1717, SE = 0.0605, 95% CI: [−0.3020, −0.0618]).

Moderation analysis: The moderation analysis of this study explored the relationship between the boundary of the unhealthy food packaging information, self-construal, and their impact on product evaluation. The results showed that the interaction between the boundary of the unhealthy food packaging information and self-construal was significant (F (1, 334) = 28.88, *p* < 0.001, η^2^ = 0.080). This suggests that the boundary of unhealthy food packaging information and self-construal interact to impact consumers’ product evaluation. For the independent participants, those in the boundary-presence condition (M _presence_ = 4.78, SD _presence_ = 1.43; F (1, 174) = 28.80, *p* < 0.001, η^2^ = 0.142) rated the cookies lower compared to those in the boundary-absence condition (M _absence_ = 5.70, SD _absence_ = 0.74). For the interdependent participants, those in the boundary-presence condition (M _presence_ = 5.22, SD _presence_ = 0.89; F (1, 160) = 5.20, *p* = 0.024, η^2^ = 0.031) rated the cookies higher compared to those in the boundary-absence condition (M _absence_ = 4.82, SD _absence_ = 1.31) (see Figure 7).

Alternative explanations: We conducted an ANOVA with control perception, product liking, and price as the dependent variables and the unhealthy food packaging information boundary as the independent variable. The results indicated that the participants’ control perception (M _absence_ = 5.21, SD _absence_ = 1. 18 vs. M _presence_ = 5.31, SD _presence_ = 1.17; F (1, 336) = 0.649, *p* = 0.421, η^2^ = 0.002), product liking (M _absence_ = 5.70, SD _absence_ = 1.28 vs. M _presence_ = 5.52, SD _presence_ = 1.39; F (1, 336) = 1.57, *p* = 0.211, η^2^ = 0.005), and price (M _absence_ = 4.51, SD _absence_ = 1.44 vs. M _presence_ = 4.71, SD _presence_ = 1.39; F (1, 336) = 1.73, *p* = 0.189, η^2^ = 0.005) were not affected by the unhealthy food packaging information boundary. Additionally, we conducted mediation analyses using 5000 bootstrap samples [82] to examine the indirect effects of control perception, product liking, and price. The mediating path of control perception (β = 0.0456, SE = 0.0566, 95% CI = [−0.0633, 0.1595]), product liking (β = −0.0864, SE = 0.0705, 95% CI = [−0.2298, 0.0467]), and price (β = 0.0345, SE = 0.0292, 95% CI = [−0.0179, 0.0984]) was not significant. These observations suggest that control perception, product liking, and price are not underlying explanatory factors.

### 5.5. Discussion

In Experiment 3, after controlling brand effects, the main effect was further confirmed. Moreover, an important boundary condition was identified regarding the influence of unhealthy food packaging information boundaries on consumer purchase intention, namely, self-construal. When consumers exhibit an independent self-construal, their purchasing intention decreases in the presence (vs. absence) of unhealthy food packaging information boundaries. Conversely, when consumers exhibit an interdependent self-construal, their purchasing intention increases in the presence (vs. absence) of unhealthy food packaging information boundaries.

## 6. General Discussion

This study examined the effect of the presence (vs. absence) of boundaries in unhealthy food packaging information on consumers’ purchasing intentions. Utilizing a progressive experimental design with various unhealthy foods, boundary manipulation methods, purchasing intention measurement techniques, and self-construal manipulation methods, this research found a negative influencing effect of the presence (vs. absence) of an unhealthy food packaging information boundary on consumer purchasing intention (Study 1 and Study 2). The feeling of constraint emerged as an intrinsic mechanism underlying this influence (Study 2). In addition, the moderating role of self-construal was confirmed (Study 3). Specifically, consumers with an independent self-construal with the presence (vs. absence) of unhealthy food packaging information boundaries had decreased purchasing intentions, whereas consumers with a dependent self-construal with the presence (vs. absence) of unhealthy food packaging information boundaries had increased purchasing intentions.

### 6.1. Theoretical Contributions

The theoretical contributions of this study are as follows.

First, it expands the research related to unhealthy food packaging information by examining the effect of the presence (vs. absence) of boundaries in unhealthy food packaging information on consumers’ purchasing intentions. Previous studies have identified several factors influencing consumers’ purchasing of unhealthy foods, such as personal habits, personality traits, socioeconomic conditions, and self-health status [29]. Some studies have also explored ways to inhibit consumers from purchasing unhealthy foods, such as exposing consumers to the odors of unhealthy foods [9] and using digital exposure techniques [12]. These existing studies have primarily focused on the influence of the consumers themselves or the consumption environment on their perceptions of unhealthy foods. Research on unhealthy food packaging has mostly addressed the information conveyed by the packaging, such as different package colors signaling different health messages [14], or the inhibitory effect of food nutrition labels on consumers’ purchasing behavior [16], with less attention given to the boundaries of food packaging information. This study confirms that the presence of boundaries in unhealthy food packaging information affects consumers’ purchasing intentions and further reveals the intrinsic mechanism by which these boundaries influence purchasing behavior. By achieving this, it enriches the current academic research on unhealthy foods and consumers’ purchasing intentions.

Second, this paper approaches the topic from the perspective of the feeling of constraint, offering a novel explanation for how the presence (vs. absence) of information boundaries on unhealthy food packaging products reduces consumers’ willingness to purchase. The feeling of constraint is an ever-present influence on consumer behavioral choices. Although research on the feeling of constraint within consumer behavior has progressed in recent years, many untested questions remain. Only a few studies have explored the impact of the feeling of constraint on consumer behavior in specific contexts. For example, the sense of financial constraint decreases consumers’ choices of green products [95] and increases their choices of high-calorie foods [96,97]. Given the significant role of the feeling of constraint in the current consumer market and its incomplete exploration, this paper attempts to elucidate the mechanism by which unhealthy food packaging information boundaries influence consumers’ purchasing intentions from the perspective of the feeling of constraint. The results provide further evidence supporting theories related to the feeling of constraint, thereby enriching the existing literature on consumer behavior.

Finally, this paper expands the understanding of self-construal. Prior research has shown that different types of self-construal orient individuals towards distinct consumer goals and social orientations, impacting their psychology and decision-making [25]. For example, consumers with different types of self-construal exhibit variations in product choice [98,99], brand preference [100,101], and more. Unlike these studies, this paper reveals that the different types of self-construal led to divergent responses to the presence (vs. absence) of boundaries in unhealthy food packaging information, thereby affecting purchasing intentions. Specifically, consumers with an independent self-construal with unhealthy food packaging information boundaries presence (vs. absence) had decreased purchasing intentions, while consumers with interdependent self-construal with unhealthy food packaging information boundaries presence (vs. absence) increase purchasing intentions. This finding extends the known effects of self-construal and enriches the existing literature on self-construal research.

### 6.2. Practical Contributions

This study offers valuable insights into the regulation and policy formulation of unhealthy foods.

This study demonstrates a reduction in consumers’ willingness to purchase these foods when there is a boundary in the packaging information of unhealthy foods. From a public management perspective, these findings provide theoretical support for relevant authorities aiming to control the consumption of unhealthy foods. When choosing food-related products, consumers’ emotions tend to change, which plays a crucial role in the final purchasing behavior [102]. Regulators can leverage this understanding by requiring unhealthy food producers to incorporate boundaries into their food packaging information. This strategy could effectively influence consumer attitudes and purchasing behavior. Implementing such packaging regulations is also more straightforward and easier than enforcing regulations in other areas of food advertising, such as digital marketing [103].

Feelings of constraint can significantly impact consumers’ willingness to purchase unhealthy foods. Beyond manipulating the boundaries of unhealthy food packaging information, regulators can employ additional strategies to evoke feelings of constraint and thereby reduce the purchasing of such foods. For example, retailers could strategically place constraint-inducing questions alongside the unhealthy food aisles. These questions would not only capture consumers’ attention but also prompt them to consider the difficulties and challenges they may encounter during the shopping process. This dual effect, drawing attention and inducing reflective thinking, could further diminish consumers’ inclination to purchase unhealthy foods.

The findings indicate that independent self-construal decreases purchasing intentions when unhealthy food packaging information has boundaries, whereas interdependent self-construal increases them. Given that consumer self-construal can be influenced, it is essential to utilize communication strategies, such as advertisements, games, and social media, to activate independent self-construal, especially for collectivist cultures, among women, or in other consumer groups characterized by a strong interdependent self-construal.

### 6.3. Research Limitations and Future Work

This paper describes the influence of unhealthy food packaging information boundaries on consumers’ purchasing intention and explores its inner mechanism and boundary conditions, which have certain theoretical value and practical significance. However, it is undeniable that there are still some limitations in this study due to the limitations of the conditions, and future research can be further expanded based on this paper.

Firstly, this study only examined one type of constraint induced by boundaries. Future research could explore other types of constraints; for example, previous research has demonstrated that the size of organic food packaging can influence consumer purchasing intentions [104]. Consequently, does the restriction of packaging size similarly affect the willingness to purchase unhealthy foods? Secondly, there may be other boundary conditions influencing the impact of unhealthy food packaging information boundaries on consumers’ purchasing intentions, which future studies need to investigate. Finally, the data for this study were collected from an online platform. Although efforts were made to create a perception of a real decision-making environment during the experiment, consumer behavior in real-world settings may differ from that in hypothetical scenarios. Therefore, whether the results of this study can be generalized to real-life situations requires further validation.

## Figures and Tables

**Figure 1 foods-13-02320-f001:**
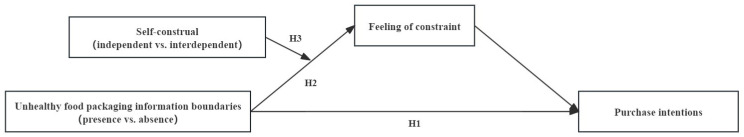
Conceptual model.

**Figure 3 foods-13-02320-f003:**
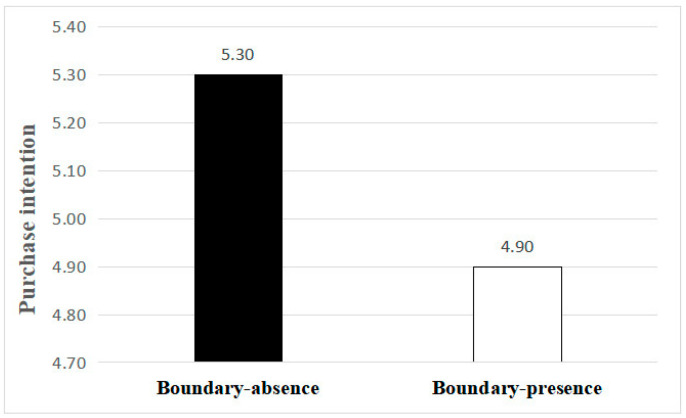
The effect of the boundary of unhealthy food packaging information on purchasing intention.

**Figure 5 foods-13-02320-f005:**
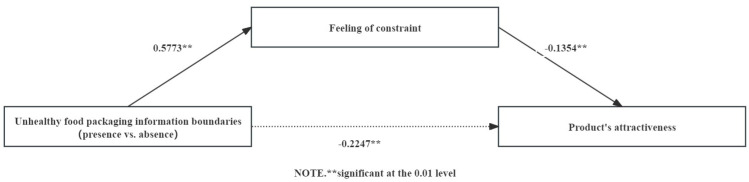
Mediating effect of feeling of constraint.

**Figure 7 foods-13-02320-f007:**
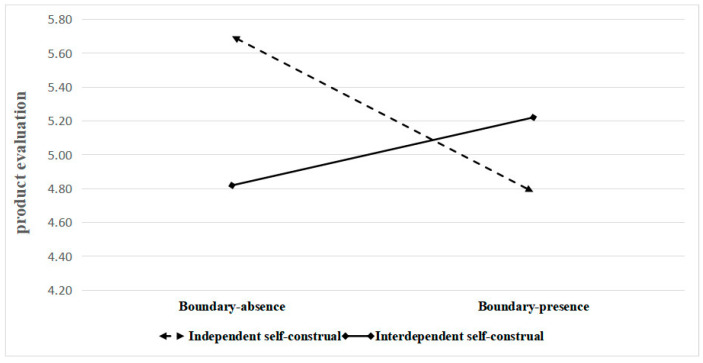
Interaction of the boundary of unhealthy food packaging information and self-construal.

## Data Availability

The data presented in this study are available on request from the corresponding author. The data are not publicly available due to the need to maintain the confidentiality of the study participants.

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
