# Peer review of "The Effect of Unhealthy Food Packaging Information Boundaries on Consumer Purchasing Intentions"

_foods, 2024, doi:10.3390/foods13152320_

Round 1

Reviewer 1 Report

Comments and Suggestions for Authors

The paper presents a very interesting and valuable study with fruitful findings on the impact of unhealthy food packaging information boundaries on consumer purchase intentions. The experimental design is robust, and the results provide significant insights into consumer behavior and policy implications. However, there are a few areas where additional details and clarifications are needed. I suggest a minor revision to address the following points:

1.       Conceptual Model Figure for H1:

The authors present a conceptual model figure for Hypothesis H2 but not for Hypothesis H1. Please include a conceptual model figure when presenting Hypothesis H1. This will help in visualizing the theoretical framework and make the presentation of the hypotheses more consistent and easier to understand.

2. Detailed Description of Experimental Procedures:

The description of the experimental procedures lacks some details regarding the respondents' interaction with the stimulus and the experimental environment.

-       Time Allowed for Stimulus Exposure: Specify how much time the respondents had to view the stimulus before answering the questions.

-       Consultation of Stimulus: Clarify whether the participants could see or consult the stimulus while answering the questions.

-       Researcher Assistance: Explain if there was any researcher present (digitally or otherwise) to assist the respondents during the experiments. Mention if the researchers could help the respondents during the answering of questions.

-       Incentives: Indicate whether the participants received any incentives for participating in the study.

3. Translation of Packaging Information:

The packaging information used in the experiments is not translated into English, limiting the understanding of non-Chinese-speaking readers.

It would be nice if authors could translate the information on the packaging into English and present this translation in the appendix. This will allow all readers to fully understand the context and content of the packaging used in the experiments.

Last, there appears to be a typographical error on line 440.

The paper makes significant contributions to the literature on unhealthy food packaging and consumer behavior. Addressing the above points will enhance the clarity and comprehensiveness of the study. I recommend a minor revision to incorporate these suggestions.

Comments on the Quality of English Language

The paper presents valuable research, but some areas exhibit awkward phrasing, grammatical errors, and inconsistencies that hinder comprehension. Improving these aspects will enhance the clarity and professionalism of the manuscript.

Author Response

Comments 1: The authors present a conceptual model figure for Hypothesis H2 but not for Hypothesis H1. Please include a conceptual model figure when presenting Hypothesis H1. This will help in visualizing the theoretical framework and make the presentation of the hypotheses more consistent and easier to understand.

Response 1: Thank you for your careful review of our manuscript and your constructive comments. We agree with this comment.

Actually, there is a conceptual model in Figure 1 of our manuscript, but the Hypotheses are not labeled. Based on your valuable feedback, we have revised and annotated the conceptual model. Specifically, we labeled the parts corresponding to H1, H2, and H3 clearly in the model diagram, facilitating readers to quickly locate each assumption's position in the model and enhancing the visual expression of the intrinsic connections and logical relationships among the assumptions. The model with these supplementary labels is presented in Figure 1, highlighted in red within the original text on page 6, lines 256-457.

Thank you again for your valuable comments and hard work, and your feedback motivates us to continue improving. We look forward to receiving your ongoing support and guidance in our subsequent research.

Comments 2: The description of the experimental procedures lacks some details regarding the respondents' interaction with the stimulus and the experimental environment.

--Time Allowed for Stimulus Exposure: Specify how much time the respondents had to view the stimulus before answering the questions.

--Consultation of Stimulus: Clarify whether the participants could see or consult the stimulus while answering the questions.

--Researcher Assistance: Explain if there was any researcher present (digitally or otherwise) to assist the respondents during the experiments. Mention if the researchers could help the respondents during the answering of questions.

--Incentives: Indicate whether the participants received any incentives for participating in the study

Response 2: Thank you very much for your valuable review comments. We agree with this comment. And we have carefully considered your suggestions. The manuscript has been revised accordingly, especially, since we have made adjustments to the experimental procedures in the three experiments, with specific modifications detailed below:

Experiment 1:

--Time Allowed for Stimulus Exposure: In section 3.3, following the description of the requirement for participants to view the information on Sam's hamburger packaging, we have added the sentence: "Participants are required to remain in this stage for 1 minute to carefully observe the stimulus".

--Consultation of Stimulus: In section 3.3, "Experimental Procedure", we added the sentence "The experiment included two parts: viewing stimuli and answering questions" to clarify that participants could not view the stimuli while responding to the questions.

--Researcher Assistance: In section 3.2, "Experimental sample and design", following the description of the sample, we have added the sentence "The entire experimental process was conducted online without any researcher assistance".

--Incentives: At the end of section 3.2, we added the sentence "After the experiment, each participant was randomly paid 5–10 yuan" to clarify that participants received compensation for their participation.

The experiment 1 modifications have been marked in red within the text, located on page 6, lines 269-271, line 277, and lines 288-289.

Experiment 2:

--Time Allowed for Stimulus Exposure: In section 4.3, same as experiment 1, following the description of the requirement for participants to view the information on chips packaging, we have added the sentence: "Participants are required to remain in this stage for 1 minute to carefully observe the stimulus".

--Consultation of Stimulus: In section 4.3, Following the description of the participants' task of viewing the stimuli, we added the sentence "After viewing the picture stimulus, participants proceeded to the next screen to answer the questions, where the stimulus image would no longer be visible".

--Researcher Assistance: In section 4.2, same as experiment 1, following the description of the sample, we have added the sentence "The entire experimental process was conducted online without any researcher assistance".

--Incentives: At the end of section 4.2, "Experimental sample and design", we also added the sentence "After the experiment, each participant was randomly paid 5–10 yuan".

The experiment 2 modifications have been marked in red within the text, located on page 8, lines 355-357, and lines 367-370.

Experiment 3:

--Time Allowed for Stimulus Exposure: In section 5.3, consistent with the preceding experiments, following the description of the requirement for participants to view the information on chocolate cookie packaging, we have added the sentence: "Participants are required to remain in this stage for 1 minute to carefully observe the stimulus".

--Consultation of Stimulus: In section 5.3, following the description of the participants' task of viewing the stimuli, we added the sentence "After viewing the picture stimulus, participants proceeded to the next screen to answer the questions, where the stimulus image would no longer be visible".

--Researcher Assistance: In section 5.2, we add this content. Consistent with the preceding experiments, the entire procedure was conducted online without researcher intervention.

--Incentives: At the end of section 5.2, "Experimental sample and design", we added the sentence " Upon completion, participants were compensated with a randomized payment of 5 to 10 yuan".

The experiment 3 modifications have been marked in red within the text, located on page 11, lines 451-453, and lines 470-472.

Thank you again for your valuable feedback.

Comments 3: The packaging information used in the experiments is not translated into English, limiting the understanding of non-Chinese-speaking readers. It would be nice if the authors could translate the information on the packaging into English and present this translation in the appendix. This will allow all readers to fully understand the context and content of the packaging used in the experiments.

Response 3: Thank you for pointing this out. We agree with this comment. Upon receiving your review comments, we conducted a comprehensive review of the packaging information discussed in the paper. To accommodate non-Chinese-speaking readers, we have translated the packaging information into English. To enhance the fluency of the translation, we have implemented the following measures.

First, we invited an expert from an English-speaking country to assist in translating the packaging information used in the experiment. During the translation process, we endeavor to preserve the accuracy and professionalism of the original text while ensuring that the translation remains comprehensible. Second, upon completion of the translation, we utilized AI software to refine the translated content. This process ensured the fluency, accuracy, and academic quality of the paper, aligning its language with international academic standards. Naturally, the final results were meticulously reviewed and manually adjusted to ensure consistency with our research intent and academic style. Finally, we have presented the translated product information in the appendix, formatted consistently with the package information. For your reference, the translated content is provided below:

The stimulus utilized in Experiment 1 was Sam's hamburger, with the corresponding English translation of its package information provided in Appendix C. The details are as follows.

--Product name: Angus Beef Cheese Burger

--Ingredients List: Angus Beef Patty (Angus beef, onion, salt, moisture-retaining agents [sodium tripolyphosphate, sodium pyrophosphate, monopotassium phosphate], white sugar, spices, leavening agent [sodium bicarbonate]), High-Gluten Wheat Flour, American Cheddar Cheese Slices (processed cheese), Drinking Water, Low-Gluten Wheat Flour, Unsalted Butter (light cream, cream), White Sugar, Vegetable Protein Liquid (drinking water, vegetable oil, pea protein, glucose, maltodextrin), Whole Egg Liquid, Malt Syrup, Dry Yeast, Salt, Trehalose, Shortening, Food Flavoring, Compound Stabilizing Enzyme Preparation (calcium sulfate, ascorbic acid, xylanase, maltogenic amylase, α-amylase, glucose oxidase, lipase, wheat flour, soybean flour), Calcium Propionate, Compound Coloring Agent (sodium starch octenyl succinate, β-carotene, sodium erythorbate, citric acid, maltodextrin, refined vegetable oil)

--Net content: 1.05 kg (6 pieces * 175 g/piece)

--Shelf life: 6 months

--Storage condition: <-18℃ frozen storage

--Reminder: After thawing, please consume within 2 days

The stimulus utilized in Experiment 2 was Oishi chips, with the corresponding English translation of its package information provided in Appendix D. The details are as follows.

--Product name: Oishi shrimp flakes

--Net content: 7g*15 7g*30

--Ingredients List: Shrimp, wheat flour, vegetable oil, starch, white sugar, edible salt

--Date of manufacture: See packaging

--Shelf life: 9 months

--Storage method: Store in a cool, dry place, away from direct sunlight

The stimulus utilized in Experiment 3 was Chocolate cookies, with the corresponding English translation of its package information provided in Appendix E. The details are as follows.

--Net content: 85g

--Shelf life: 9 months

--Origin: China     

--Food Production License No. SC11332057100228

--Ingredients List: Wheat Flour, White Sugar, Edible Vegetable Oil, Cocoa Butter Substitute Chocolate (contains Soy Lecithin), Cocoa Powder, Food Additives (Ammonium Bicarbonate, Sodium Bicarbonate, Disodium Pyrophosphate, Diacetyl Tartaric Acid Esters of Mono- and Diglycerides), Starch, Edible Salt, Edible Flavorings.

--Allergen information: Contains wheat, soy products, and dairy products. This production line also processes products containing sesame, egg products, and other grain products containing gluten

--Storage conditions: Store in a cool and dry place, away from direct sunlight

The above modifications have been marked in red within the text, located on pages 18-20, lines 704-760. Finally, we would like to thank you again for your kind guidance and valuable suggestions. Your feedback is not only an affirmation of our work but also an important motivation for our future research. If you have any additional suggestions or need further clarification while reviewing the revised manuscript, please feel free to contact us. We look forward to maintaining continuous communication with you.

Comments 4: There appears to be a typographical error on line 440.

Response 4: Thank you very much for your careful review of our manuscript. We agree with this comment.

As soon as we received your comments, we scrutinized line 440, as well as the entire manuscript, to ensure there were no further typographical errors. We are well aware that academic publishing is a pursuit of excellence, and any minor errors may affect readers' understanding and trust. Therefore, we especially appreciate your valuable suggestions and patient guidance.

The revised text, indicated by redlining, can be found on page 3, lines 106-107; page 6, lines 258-260; page 8, lines 342-344; page 10, lines 435-437. If you have any other suggestions or need further information while reviewing the revised manuscript, please feel free to contact us. We are always willing to provide the necessary answers.

Comments 5: The paper presents valuable research, but some areas exhibit awkward phrasing, grammatical errors, and inconsistencies that hinder comprehension. Improving these aspects will enhance the clarity and professionalism of the manuscript.

Response 5: Thank you for pointing this out. We agree with this comment. To address these issues, we enlisted the assistance of native-speaking experts and conducted a comprehensive review and revision of our manuscript. Subsequently, we utilized artificial intelligence software for further refinement. Additionally, we conducted a thorough proofreading of the entire article.

Throughout these three stages of review, we identified several issues, such as on page 1, line 16 of the original text, where the word was originally expressed as " presence (vs. absence)". Following the review, we have revised it to " present (vs. absent)". For instance, on page 1, lines 32-33 of the original text, where the sentence was originally expressed as " Consequently, numerous scholars have investigated the reasons for the consumption of unhealthy foods" Following the review, we have revised it to " Consequently, numerous scholars have investigated the reasons for consuming unhealthy foods ".

Similarly, we identified other areas with inappropriate English expressions, which we have revised and marked in red. These specific revisions are located on page 1, lines 9-10, 33, 43; page 2, lines 44, 46, 55, 58-59, 61, 68, 74, 85; page 3, lines 142, 144; page 4, lines 150, 181, 186, 193; page 5, line 215; page 6, line 252, 254, 272, 291, 294; page 7, line 295-296; page 8, line 339,348, 358; page 10, line 412, 429-430; page 11, line 443, 445, 474; page 14, line 596; page 15, line 645, 652-653, 656.

We appreciate your understanding and patience as we strive to improve the clarity and accuracy of our research article.

Reviewer 2 Report

Comments and Suggestions for Authors

In the abstract, you have to specify (adding values) your results. What can we conclude with this study and what are their practical implications and future perspectives? Please, highlight this in the abstract as well.

Lines 54-61: References are missing.

Section 2 is too exhaustive. If we consider the section 1 and 2 as the background of this manuscript this is very, very extensive. In my opinion should reduce it and bring it to the essentials.

The titles of sections 3, 4, 5, and 6 are not adequate. You should structure your manuscript in Introduction; Materials and Methods; Results and Discussion (you may separate these 2 sections); and Conclusions. In its current form is very hard to follow the steps you took to carry out your investigations.

The quality of the figures has to be improved.

After you reorganize your manuscript, I suggest you make a new submission. In its current form is not acceptable.

Author Response

Comments 1: In the abstract, you have to specify (adding values) your results. What can we conclude with this study and what are their practical implications and future perspectives? Please, highlight this in the abstract as well.

Response 1: Thank you for pointing this out. We agree with this comment. For the question you raised, we have revised it as follows.

In the original abstract, we only described the research results and did not include the conclusion of this study and what are their practical implications and future perspectives. Regarding the practical implications and future research directions of this study, we have summarized our findings in the abstract as follows: This paper reveals the psychological mechanism and boundary conditions of unhealthy food packaging information boundary affecting consumers' purchase intention and provides practical inspiration for government policy-making related to unhealthy food packaging.

We have marked the modifications in red in the original text, located on lines 9-20 of the first page of this article. Finally, thank you again for providing us with such constructive perspectives, which have made our abstract richer and easier to understand.

Comments 2: Lines 54-61: References are missing

Response 2: Thank you for pointing this out. We agree with this comment. We have added the missing references in lines 54-61 to enhance clarity. Specifically, as follows.

We concluded that in studies dedicated to unhealthy food packaging, the focus is mainly on the content of the message, such as color and wording. However, no references were provided for these examples. We have now addressed this by citing reference 14 for color and reference 15 for text. The insights regarding the impact of unhealthy food packaging colors on consumer perception were drawn from a 2015 study titled "Eat with your eyes: Package color influences the expectation of food taste and healthiness moderated by external eating.". The findings related to the influence of unhealthy food packaging text were derived from a 2023 study titled "The length of brand names influences the expectation of healthiness in foods and preference for healthy foods."

We have marked this section in red, and the update is located on lines 60-61 of the second paragraph on page 2 of this paper. Finally, we would like to thank you again for helping us identify this error, which has made our article more rigorous and objective.

Comments 3: Section 2 is too exhaustive. If we consider the section 1 and 2 as the background of this manuscript this is very, very extensive. In my opinion should reduce it and bring it to the essentials.

Response 3: Thank you very much for your careful review and valuable comments on our manuscript. We agree with this comment. Your feedback provides crucial guidance for refining and improving the manuscript's content. In response to your concern about Section 2 being too detailed, we have made substantial revisions to ensure the completeness of the information while making the content more concise and the core themes more prominent. Specifically, we have taken the following measures to optimize the structure and content of Section 2:

Adjustment of Subsection 2.1: We deeply understand your request for streamlining and have carefully consolidated the original four paragraphs into two. In the first paragraph, we focus on the basic definition of unhealthy food and its related research. By culling common examples of unhealthy food and nutrient indicators, we have retained only some essential examples to help readers quickly grasp the core features of unhealthy food. Additionally, we have rewritten the section on the causes of unhealthy food consumption and inhibition strategies, aiming to highlight the most critical conclusions for readers' convenience and understanding. The second paragraph then focuses on the current state of research on unhealthy food packaging, summarizing key points to lay the foundation for subsequent discussions.

Adjustment of Subsection 2.2: Following your suggestions, we have merged the original two paragraphs into one and carefully polished the sentences. By deleting repetitive content and simplifying sentence structures, we have ensured that the information is conveyed directly and clearly. Meanwhile, for the citation of reference 48, we have streamlined the example and retained only its most representative conclusions, thus maintaining the conciseness of the article without sacrificing clarity

Adjustment of Subsection 2.3: We have adopted your suggestion and deleted the sentence, "Some scholars expanded on this idea by manipulating physical spatial boundaries through the height of the ceiling." We recognize that this sentence adds contextual information but does not directly relate to the core argument of this subsection. Therefore, we have removed this part to make the focus of subsection 2.3 clearer and the conclusion more prominent.

Adjustment of Subsection 2.4: Under your guidance, we have restated the definition of self-construal. By shortening the text and strengthening the contrast, we have made a more intuitive distinction between independent and interdependent self-construal.

Finally, thank you again for your support and guidance on our work. We believe that these revisions make the manuscript more concise, structurally sound, and logically clearer. The revised text, indicated by redlining, can be found on pages 3-6, lines 108-254. We look forward to your further review and are prepared to make additional improvements based on your new feedback.

Comments 4: The titles of sections 3, 4, 5, and 6 are not adequate. You should structure your manuscript in Introduction; Materials and Methods; Results and Discussion (you may separate these 2 sections); and Conclusions. In its current form is very hard to follow the steps you took to carry out your investigations.

Response 4: Thank you very much for your detailed review and valuable comments on our paper. We agree with this comment. We have discussed and deeply reflected on the issues you raised and made the following specific modifications:

First, regarding the insufficiency of the title you mentioned, we immediately made adjustments by adding the title of section 3 as “Validating the Effect of Unhealthy Food Packaging Information Boundaries on Purchase Intentions”, the title of section 4 as “The Mediating Effect of Feeling of Constraint on the Relationship Between Unhealthy Food Packaging Information Boundaries and Purchase Intentions”, and the title of section 5 as “The Moderating Effect of Self-Construal on the Relationship Between Unhealthy Food Packaging Information Boundaries and Purchase Intentions”.

Second, in response to your suggestion to structure the manuscript, we made the following adjustments after careful consideration and many references. We constructed the manuscript as an Introduction, Literature Review and Research Hypotheses, Experiment 1 - Experiment 3, and General Discussion, totaling six sections. We made this adjustment for the following reasons:

The research methodology used in this manuscript was experimental. The Experiment 1-3 sections present different experimental designs and results, respectively. We were concerned that separating the procedure and the results would affect reading fluency. However, we are also aware of the potential shortcomings of the existing framework. To address this, we examined articles published in recent years in foods journals that employed the experimental approach as a research method, such as " Catarci, D.; Laasner Vogt, L.; Reijnen, E. Online Food Choices: When Do "Recommended By" Labels Work? Foods. 2024, 13, 928." and " Liang, S.; Qin, L.; Zhang, M.; Chu, Y.; Teng, L.; He, L. Win Big with Small: The Influence of Organic Food Packaging Size on Purchase Intention. Foods. 2022, 11, 2494.". We have referred to these published articles to reconstruct the framework of this paper. Under the new structure, our paper introduces the research background and problem through the Introduction section. The Literature Review and Research Hypotheses section elaborates on the relevant theoretical foundations and hypotheses. The Experiment 1-3 sections show different experimental designs and results, and finally, the experimental results are comprehensively analyzed and discussed in the General Discussion section. We hope that this structure allows readers to follow our thoughts and gradually understand the research content in depth.

The revised text, indicated by redlining, can be found on page 3, line 106; page 6, lines 257-258; page 8, lines 341-342; page 10, lines 434-435. Thank you again for your patient review and valuable suggestions on our thesis. We look forward to further improving the thesis in subsequent revisions and hope to eventually receive your approval and support. Please feel free to let us know if you have any further revisions or suggestions.

Comments 5: The quality of the figures has to be improved.

Response 5: Thank you for pointing this out. We agree with this comment. We are well aware of the importance of image quality for clearly expressing research content in the paper. Therefore, we deeply value your feedback. Upon receiving your review comments, we immediately reviewed the images in the paper and acknowledged the deficiencies you highlighted. To enhance the overall quality of the paper and ensure the accurate communication of the research results, we have taken the following measures to improve the quality of the images.

Firstly, we used professional software to further optimize the images by adjusting contrast, brightness, and sharpness to improve visual effect and readability. Next, we reproduced the images using higher-resolution software to ensure that each image reached at least 300 dpi, meeting the standard clarity required for publication. Finally, after completing these improvements, our team conducted an internal review and proofread the images again to ensure they were of optimal quality and met the journal's submission requirements.

The revised text, indicated by redlining, can be found on page 9, lines 382-384; page 12, lines 486-488. Thank you again for your patience and valuable suggestions, which have significantly improved the quality of our paper. If you have any further questions or need additional materials, please feel free to let us know.

Round 2

Reviewer 2 Report

Comments and Suggestions for Authors

In the abstract, you have to specify your results by adding values. I have already commented on this earlier. Highlight with values the main results of your experiments.

Author Response

Comments 1: In the abstract, you have to specify your results by adding values. I have already commented on this earlier. Highlight with values the main results of your experiments.

Response 1: Thank you for pointing this out. We agree with this comment. For the question you raised, we have revised it as follows.

In the original abstract, we only described the research results and did not include specific values, which we have now added in detail. Experiment 1 primarily verifies Hypothesis 1, which posits that the presence (vs. absence) of packaging information boundaries on unhealthy foods has a negative impact on consumers' purchase intentions. This effect was found to be significant (p = 0.040). Experiment 2 tests Hypothesis 2, asserting that the feeling of constraint mediates this effect. Using a mediation model (Model 4, with 5000 bootstrapping samples), we confirmed this hypothesis, and the results were significant (β = -0.078, CI: [-0.1911, -0.0111]). Experiment 3 examines Hypothesis 3, which investigates the role of self-construal. Specifically, consumers with an independent self-construal exhibit reduced purchase intentions when unhealthy food packaging information boundaries are present (vs. absent) (p<0.001), whereas those with an interdependent self-construal show increased purchase intentions under the same condition (p=0.024).

We have marked the modifications in red in the original text, located on lines 14-18 of the first page of this article. Finally, thank you again for providing us with such constructive perspectives, which have made our abstract richer and easier to understand.